Effect of seedling size on post-planting growth and survival of five Mexican Pinus species and their hybrids

Ponce-Figueroa José Alberto 1
Antúnez Pablo 2
http://orcid.org/0000-0002-3284-422X Hernández-Díaz José Ciro 3
http://orcid.org/0000-0002-2954-535X Prieto-Ruíz José Ángel 4
Carrillo-Parra Artemio 3
López-Serrano Pablito Marcelo 3
http://orcid.org/0000-0002-2341-5458 Wehenkel Christian 3 wehenkel@ujed.mx
1 Maestría Institucional en Ciencias Agropecuarias y Forestales (MICAF), Universidad Juárez del Estado de Durango , Durango, Durango , Mexico
2 División de Estudios de Posgrado, Instituto de Estudios Ambientales, Universidad de la Sierra Juárez , Oaxaca, Oaxaca , Mexico
3 Instituto de Silvicultura e Industria de la Madera, Universidad Juárez del Estado de Durango , Durango, Durango , Mexico
4 Facultad de Ciencias Forestales y Ambientales, Universidad Juárez del Estado de Durango , Durango, Durango , Mexico
Arena Carmen
Electronic publication date: 2024 Dec 20
Publication date: 2024
Volume: 12
Electronic Location ID: e18725
Received 2024 Sep 5; Accepted 2024 Nov 26
Copyright: © 2024 Ponce-Figueroa et al.
Copyright year: 2024
Copyright holder: Ponce-Figueroa et al.
License: This is an open access article distributed under the terms of the Creative Commons Attribution License, which permits unrestricted use, distribution, reproduction and adaptation in any medium and for any purpose provided that it is properly attributed. For attribution, the original author(s), title, publication source (PeerJ) and either DOI or URL of the article must be cited.
License URL: https://creativecommons.org/licenses/by/4.0/

Keywords: Seedling quality, Seed provenance, Random Forest, Unbiased conditional coefficient of determination, Seedling dimension, Microhabitat, Environmental heterogeneity, Trial field

Funding: Mexican National Forestry Commission (CONAFOR) Mexican Council of Humanity, Science and Technology (CONAHCyT) 1179335 Durango State Council of Science and Technology (COCyTED) 120505 This study was supported by joint funding from the Mexican National Forestry Commission (CONAFOR), the Mexican Council of Humanity, Science and Technology (CONAHCyT) - Finance Code 1179335 and Durango State Council of Science and Technology (COCyTED) - Finance Code 120505. The funders had no role in study design, data collection and analysis, decision to publish, or preparation of the manuscript.

==============================
Background

Seedling growth and survival depend on seedling quality. However, there is no experimental evidence showing that the seedling dimensions of the abundant, economically important and widely distributed tree species Pinus arizonica, P. durangensis, P. engelmannii, P. leiophylla, and P. teocote and their hybrids effectively improve survival and growth in reforestations and plantations in Mexico. Therefore, the aim was to evaluate the influence of initial morphological parameters of 2,007 nursery seedlings of these species and their hybrids on their growth and survival 44 months after planting in the Sierra Madre Occidental, Mexico.

Methods

Spearman’s coefficient (rs) and the unbiased conditional pseudo coefficient of determination (R2c) between each specific predictor and each response variable and their 95% confidence interval (CI95%) were determined using Random Forest, generalized linear model, and bootstrapping. By bootstrapping, the potential environmental heterogeneity inside the trial fields and its impact on the results were also quantified.

Results

Among the studied species and their hybrids moderate correlations were observed between the nursery seedling dimensions and the plant dimensions 44 months after planting. However, only weak significant correlations were found between survival rate (SR) and height (H) (rs = 0.10) and between SR and robustness index (HRCD) both before planting (rs = 0.06). Also, weak significant R2c values of the seedlings RCD, H and HRCD were detected with respect to the corresponding RCD, H and SR 44 months after planting, respectively. Furthermore, the predictor variable “seed provenance” (with 23 provenances) significantly explained the variation in the post-planting RCD, H and SR of the seedlings, with R2c values ranging from 0.10 to 0.15. The low width of the CI95% shows that the environmental conditions in the trial fields were quite homogeneous.

Discussion

The results also show that the inclusion of “confounding” variables in the statistical analysis of the study was crucial. Important factors to explain this low association could be the strong damage observed caused by pocket gopher, the typically low winter-spring precipitation in both field trials and adaptation factors. The study findings provide preliminary insights and information aimed at helping to design more appropriate standards for nurseries.

Introduction

Forest ecosystems and their structural characteristics are affected by anthropogenic disturbance and extreme climate variations (Bellard et al., 2012; Franklin et al., 2016; Danneyrolles et al., 2019). Most natural forests have undergone fragmentation, leading to a reduction in the amount and quality of the environmental services they supply (Mitchell et al., 2015). In Mexico, a total area of 4.2 million hectares of forest land, representing 6.2% of the total forest cover, was deforested in the period 2001–2021 (CONAFOR, 2022). In response, both federal and state governments have taken actions to reverse the reduction in forest cover, through reforestation programs (CONAFOR, 2015).

The genus Pinus is the most widely used in Mexico to restore forest areas for ecological, economic and social reasons. Several species of this genus are valuable for timber, fuel and other resources (Rzedowski, 2006; Jin et al., 2021). The species Pinus arizonica Engelm., P. durangensis Ehren., P. engelmannii Carr., P. leiophylla Schl. & Cham., and P. teocote Schiede ex Schltdl. are abundant, economically important and widely distributed throughout northern Mexico (Hernández-Velasco et al., 2021). Furthermore, P. arizonica, P. engelmannii, and P. leiophylla also occur naturally in the southwest of the USA (Haire et al., 2022). Hybrids have been reported in these tree species (Hernández-Velasco et al., 2021). However, differences in growth and survival between these hybrids and their pure species parents were hardly checked in field trails (Sánchez-Hernández et al., 2022).

Seedling survival rate in reforestation programs in Mexico is extremely low. Mexal, Cuevas-Rangel & Landis (2008) reported that survival rates of five conifer species in Central Mexico in 1995 ranged from 15% to 86% after 2 years. Seedling quality was the cause of mortality in 0–10% of cases, depending on the species. However, data from across Mexico, compiled by Prieto-Ruíz et al. (2016) for the period 2006–2014, revealed an average mortality rate of 57% in the first year of reforestation (43% in the state of Durango). Up to 9.7% of the mortality was attributed to poor quality of the seedlings. Barrera-Ramírez, López-Aguillón & Muñoz-Flores (2018) reported that the percentage of conifer mortality at national level fluctuated between 40% and 70% between 2007 and 2017, in only the first year after plantation establishment.

It is commonly accepted that success on seedling growth and survival in the plantation field greatly depends on the quality of the seedlings (especially in nursery seedlings) (Tinus, 1974). Grossnickle & MacDonald (2018) reviewed at least 171 studies on the effects of seedlings traits on plant growth (root or shoot), observing that the use of high quality seedlings can increase the chances of successful plant establishment and growth.

Larger seedlings generally have an advantage in terms of initial growth and survival due to their larger root systems and more developed tissues, which initially can access more nutrients and water (Dierauf & Garner, 1996; Garau et al., 2009). Moreover, the robustness of the seedlings seems to affect the plant performance in the field (Li et al., 2011).

However, there is no experimental evidence showing that the seedling dimensions of the economically important tree species Pinus arizonica, P. durangensis, P. engelmannii, P. leiophylla, P. teocote and their hybrids have effect on their survival and growth in reforestations and plantations in Mexico. In order to contribute to this knowledge, in the present research we hypothesized that the growth and the survival rate in the field of these five species, are also likely to be enhanced by using nursery seedlings with larger dimensions, as has been reported for other species (Dey & Parker, 1997; Prieto-Ruíz et al., 2007; Tsakaldimi, Ganatsas & Jacobs, 2013). Therefore, the aim of this work was to evaluate the influence that the initial morphological parameters (root collar diameter and height) of nursery seedlings (planted at the age of 15 months) of these five pine species and their hybrids, had on their growth and survival at 44 months after planting.

Materials and Methods

Study area

In 2016, seeds from “pure” parental trees and “hybrid” parental trees of Pinus arizonica (PA), P. durangensis (PD), P. engelmannii (PE), P. leiophylla (PL) and P. teocote (PT) were collected from 23 seed stands (two provenances of PA, three of PD, seven of PE, two of PL, and nine of PT) in the municipality of Santiago Papasquiaro, Durango, Mexico (Tables S1 and S2). In April 2017, 3,465 seeds from these seed stands were sown in a completely random manner (i.e., an unbalanced completely randomized design). Round polypropylene containers of 162 mL were used, placed on growing beds in a forest nursery (i.e., under indoor conditions), located also in the municipality of Santiago Papasquiaro. The substrate comprised 50% of the base mixture (28% Peat moss or Canadian peat, 10% Agrolite, 12% Vermiculite) and 50% of composted pine bark with a 3 mL granulometry. The origin of each seed was known, including species affiliation, provenance, mother tree position in the seed stand and the hybrid or pure condition of its mother tree. In July 2018, a total of 2,007 seedlings grown for 15 months in the nursery were used to establish the two field trials. Of these, 965 seedlings were grown from seeds collected from pure species trees. The remaining 1,042 seedlings came from hybrid trees that Hernández-Velasco et al. (2021) had earlier identified by AFLP analysis.

One field trial was located in the Ejido Ciénega de Salpica el Agua, in the “Mesa Alta” site (25.06 N, −105.77 W and elevation of 2,710 m). The other was in the Ejido Laguna de La Chaparra, in the “Mesa Seca” site (25.12 N, −105.70 Wand elevation of 2,610 m). Both sites include pine-oak forests and belong to the municipality of Santiago Papasquiaro, Durango, Mexico). Each trial was conducted in a one-hectare plot, protected against cattle and white-tailed deer by a 1.8 m high fence. Seedlings were planted 2 × 2 m apart. Regardless of the species, provenance and type (pure or hybrid), each seedling was randomly included in both trials. In the period 1991–2021, the mean annual temperature was 11.5 °C in Mesa Seca and 10.6 °C in Mesa Alta; the mean annual precipitation was 803 mm in Mesa Seca and 903 mm in Mesa Alta. Around 85% of this annual precipitation fell during the growing season from June to October. Typically, therefore, there were seven months of dry time per year. Furthermore, it was particularly dry in the November–June 2018/2019 and 2020/2021 periods (<100 mm precipitation in each case) (CONAGUA, 2024). The soil traits varied between the two sites (sandy clay loam vs. loam; 5.88 vs. 5.25 pH; 10.99 vs. 5.35 CEC (meq/100 g)) (for further details, see Sánchez-Hernández et al., 2022). The soil traits varied between the two sites (sandy clay loam vs. loam; 5.88 vs. 5.25 pH; 12.32 vs. 7.39 nitrogen (N-NO3, kg/ha); 9.66 vs. 7.39 phosphorus (ppm); 220 vs. 116 potassium (ppm); 198 vs. 114 magnesium (ppm); 2.06 vs. 4.12 Zinc (ppm); 1,698 vs. 774 CaCO3 (ppm); 10.99 vs. 5.35 CEC (meq/100 g)) (for further details, see Sánchez-Hernández et al., 2022). The forest administrations of the ejidos “Ciénega de Salpica el Agua” and “Laguna de la Chaparra”, municipality of Santiago Papasquiaro, state of Durango, México (1005) (Engineer Fernando Salazar Jiménez) issued the field permits (field permit approval numbers: 012018 and 022018).

Data collection

Fifteen months after the seeds were sown in the nursery, we measured the root collar diameter (RCD, mm) with a digital caliper and the height from base to tip (H, cm) with a one-meter steel ruler. We then calculated the robustness index (HRCD, cm/mm2) by dividing H by RCD2. This index indicates the plant resistance to wind and its survival and growth potential in dry locations (Dickson, Leaf & Hosner, 1960). Each seedling was labelled with a unique identification code (including species–seed stand–mother tree number–Hybrid/Pure) along with RCD and H, which was first marked on the container and then on an aluminum tag attached to the seedling. This allowed monitoring and direct comparison of the dimensions of each seedling obtained from each mother tree at different times.

The second measurement was made in March 2022, i.e., 44 months after planting. The RCD and H of the surviving seedlings were re-measured and used to calculate the HRCD. The survival rate (SR) was also determined by evaluating each seedling (alive seedling = 1, dead seedling = 0), from each pure species and each hybrid mother tree (Table 1).

Table 1 Descriptive statistics of the seedlings of the five pure Pinus species and of their hybrid seedlings.

Descriptive statistics of the seedlings of the five pure Pinus species and of their hybrid seedlings, in each field trial and both trials together, 15 months after sowing in a nursery and 44 months after establishment of the field trials (March 2022).

		Mesa Alta	Mesa Seca	Both trials together	
		Mean	Min	Max	Mean	Min	Max	Mean	
Group	Variable	Pure	hybrid	Pure	hybrid	Pure	hybrid	Pure	hybrid	Pure	hybrid	Pure	hybrid	Pure	hybrid	
PA	RCD (mm)	7.2	6.9	2.7	2.7	9.9	8.8	6	6	3.4	6	8.5	6	7.1	6.8	
RCD3 (mm)	9.6	13.1	9.6	12.5	9.6	13.6	11	21.5	11	21.5	11	21.5	10.3	15.9	
H (cm)	9.9	10.6	3	3	16	16	12.5	16	12	16	13	16	10.1	11	
H3 (cm)	30	22.2	30	18	30	26.4	18.3	16	18.3	16	18.3	16	24.2	20.1	
N	26	12					2	1					28	13	
N 3	1	2					1	1					2	3	
SR	0.04	0.17					0.5	1					0.07	0.23	
PD	RCD (mm)	7.3	7.9	2.3	2.7	10	12.6	6.1	5.8	3.5	2	10	9.9	6.7	6.4	
RCD3 (mm)	11.5	11.8	7.4	7	16.51	20	15.3	15.2	9	8.2	27.2	35.1	13.7	14.5	
H (cm)	9.3	9.1	4	3	14	17	14.3	14.5	6	5	24	28	11.8	12.8	
H3 (cm)	31.4	34.2	7.6	8.9	50.2	55	36.9	32.7	11.2	9.8	76.5	80.6	34.6	33	
N	50	49					50	103					100	152	
N 3	14	15					20	58					34	73	
SR	0.28	0.31					0.4	0.56					0.34	0.48	
PE	RCD (mm)	7.4	7.5	2.3	2.7	12.6	12.6	7.5	7.4	3.1	2.3	14.6	12.9	7.4	7.4	
RCD3 (mm)	20.3	20.4	7.5	6.4	38.5	47.4	23.8	24.5	7.2	5.7	46.5	44.9	22.9	23.6	
H (cm)	9.2	9.7	3	3	16	16	9.7	9.5	3	3	30	17	9.5	9.6	
H3 (cm)	14.8	16.7	4.5	4.4	35.6	96.3	17.5	16.6	4.3	4.4	74	62	16.7	16.6	
N	243	211					336	289					579	500	
N 3	79	56					206	205					285	261	
SR	0.33	0.27					0.61	0.71					0.49	0.52	
PL	RCD (mm)	7.6	7.2	4.6	5.2	10.6	11	6.5	7.1	3.1	3.6	14	15.5	7	7.1	
RCD3 (mm)	22.5	19.3	16.6	11.7	29.3	28.7	17.8	20.1	9.4	9.5	29.2	30	19	19.9	
H (cm)	10.5	8.8	4	4	16	15.6	15.8	16	6	5	26	29	13.6	13.6	
H3 (cm)	40.8	30.8	22.6	20	54	45	34.8	33.3	19.6	19.4	47.3	50.8	36.3	32.6	
N	19	22					26	43					45	65	
N 3	4	7					12	18					16	25	
SR	0.21	0.32					0.46	0.42					0.36	0.38	
PT	RCD (mm)	7.6	7.5	2.7	1	12.9	12.9	5.9	6	1.9	3.1	15.1	15.2	6.6	6.7	
RCD3 (mm)	16.4	16.2	8.6	4.8	30.7	26.9	16.4	17.1	4.4	6.9	29.1	41.2	16.4	16.8	
H (cm)	9.6	9.5	3	3	16	17	13.6	14.4	4	4	30	28	12	12.2	
H3 (cm)	36.5	34.3	7.8	13.3	63.4	62.2	35.2	31.4	8	10.8	81	58	35.5	32.3	
N	86	137					127	175					213	312	
N 3	23	45					74	110					97	155	
SR	0.27	0.33					0.58	0.63					0.46	0.5	
Note:

Min, minimum; Max, maximum; RCD, root collar diameter (mm) before planting in the field trials (Mesa Alta and Mesa Seca), RCD3 = root collar diameter (mm) after 44 months of planting in the trials, H = height (cm) before planting in the trials, H3 = height (cm) after 44 months of planting in the trials, N = number of Pinus seedlings planted in the trials, N3 = number of surviving Pinus seedlings after 44 months of planted in the trials, SR = survival rate after 44 months, PA = Pinus arizonica, PD = P. durangensis, PE = P. engelmannii, PL = P. leiophylla, and PT = P. teocote.

The completely randomized spatial location of seedlings both in the nursery and in the two field trials, ensured that the average seedling from each mother tree and seed provenance had the same probability of being exposed to each environmental condition, from nursery sowing until the accumulated age of 59 months. Given that the seedlings were reallocated (re-randomized again using an unbalanced completely randomized design) before planting in the two field trials. Besides the seedlings at the edge of each trial were more than one tree height away from neighboring tree stands, edge effects on the seedling were marginal. However, on such large trials (1 ha) the site conditions were not expected to be homogeneous.

Data analysis

Spearman’s coefficient was used to determine the possible correlation (rs) between the RCD, H and HRCD of the seedlings before planting in the two field trials and the same variables plus SR of the seedlings 44 months after planting in the field trials, for each tree species and their hybrids, for each trial and both trials together. The correlations, their significances (p) (α = 0.05) and 95% confidence intervals (CI95%) were estimated using the cor.test function and boot package (Canty & Ripley, 2024) in R software (ver. 4.1.2, R Core Team, 2021). Bootstrapping was performed with subsamples of 50 seedlings and 1,000,000 iterations. The aim was accurately estimating the influence that the original dimensions of RCD, H and HRCD at the age of 15 months in the nursery may have on the response variables (survival rate, RCD3 and H3, 44 months after planting), while mitigating the effects of confounding variables (Wehenkel, 2020). With that purpose, an importance assessment of the predictor variables, including the original RCD, H and HRCD and also “tree species and its hybrids” (10 classes) (called species), “seed provenance” (23 classes) (called provenance) and “field trial” (two classes), was carried out, using a nested design with “seed provenance” nested to “tree species and its hybrids”. The same procedure was also performed with the predictor variable “seedling of pure or hybrid parents” (two classes) in exchange for the variable “seed provenance”.

The unbiased conditional (pseudo) coefficient of determination (R2c) between each specific predictor and each response variable, and also its 95% confidence interval (CI95%) (subsamples of 50 seedlings and 10,000 iterations) were determined using Random Forest (RF) for regression and classification (Breiman, 2001). We also applied traditional Generalized linear model (GLM) (Hardin & Hilbe, 2007) classification (for SR) and regression models (for RCD3 and H3). The randomForest package (Liaw & Wiener, 2002), rsq package (Zhang, 2023) and boot package (Canty & Ripley, 2024) in R (R Core Team, 2021) were utilized.

In addition, the potential environmental heterogeneity inside the trial fields attributable to variation in the micro-environmental conditions was also quantified. This was achieved by randomly calculating the correlations and running the models even in small subareas of the trial fields using the bootstrap procedure described above (i.e., many subsamples of 50 seedlings were taken from different subareas of about 200 m2) (Horowitz, 2001). The magnitude of such heterogeneity is reflected in the width of the CI95% of the study results, which depends on the level of microenvironment-induced variation. This variation can also include differences in human activities on the trial fields, such as land preparation and planting itself.

Comparison of RF and GLM models with and without these specific predictors (i.e., confounding predictor variables) ensures robust assessment of their importance and ultimately contributes to the accurate prediction of the response variables (i.e., R2c = R2 with all predictors including the specific predictor-R2 with all predictors excluding the specific predictor). As RCD and H were not strongly correlated with each other (rs < 0.7), but HRCD was often strongly associated with RCD or H, the RF and GLM models were constructed with both RCD and H and the other three variables “tree species and their hybrids”, “seed provenance” and “field trial” together (but without HRCD) and, also with either RCD, H or HRCD alone but together with the other three variables. For comparative purposes, additionally, biased marginal R2 between each specific predictor and each response variable without mitigating the effects of confounding (R2m) was measured, using classification (for SR) and regression (for RCD3 and H3).

In order to see clearer trends in seedling traits, we also created charts in which we visualized the relationships between classes of RCD (2 mm class width) and H classes (2 cm class width) before planting (at 15 months old) and 44 months after planting in the two field trials, across all species and their hybrids and grouped by pine species and their hybrids. Further statistical analyses were omitted because the comparison between classes, i.e., between averages, always includes an artificial improvement of the correlations.

Results

Across all species and their hybrids, there was a correlation between seedling root collar diameter before planting and 44 months after planting (RCD vs. RCD3) (rs = 0.29, CI95% = [0.23–0.35], p < 0.00001) as well as between height of the seedlings before planting and after 44 months of planting (H vs. H3) (rs = 0.40, CI95% = [0.33–0.44], p < 0.00001). However, there was only a weak, although significant, correlation between H and SR (rs = 0.10, CI95% [0.06–0.15], p = 0.000013) and also between the robustness index (HRCD) and SR (rs = 0.06, CI95% [0.02–0.11], p = 0.007).

When the five species and their hybrids were grouped together, the RCD before planting in the two field trials was often significantly and positively correlated with the RCD3 of the seedlings, but only slightly correlated with the H and SR 44 months after planting. By contrast, in the group of seedlings derived from pure P. teocote (PT-P), there was a significant negative correlation between RCD and SR (rs = −0.20, p < 0.05). The H of the seedlings of the five pine species and their hybrids before planting was sometimes significantly and positively correlated with H3 and SR of the plants, in both field trials after 44 months. In some of the studied pine species and their hybrids, 44 months after planting, HRCD was significantly and negatively correlated with RCD3 and significant positively correlated with SR. Nevertheless, there was a high degree of variation in the magnitude of the correlation between the variables (Fig. 1, Table S3).

Figure 1 Correlation between the root collar diameter, height, robustness index of seedlings before planting in 2018 and RCD, H and the survival rate of the same seedlings after planting.

Correlation (rs) between the root collar diameter (RCD, mm), height (H, cm), robustness index (HRCD, cm/mm2) of seedlings before planting in 2018 and RCD3, H3 and the survival rate (SR) of the same seedlings 44 months after planting in the field trials, for each pine pure species (-P) and their hybrids (−H). Asterisks indicate level of statistical significance: *p < 0.05, **p < 0.01, ***p < 0.001

Across all species and their hybrids under study, there were correlations between the seedling root collar diameter before planting and 44 months after planting (RCD vs. RCD3) (rs = 0.29, CI95% [0.23–0.35], p < 0.00001) as well as between height of the seedlings before planting and 44 months after planting (H vs. H3) (rs = 0.40, CI95% [0.33–0.44], p < 0.00001). But there was only a weak, although significant, association between H and SR (rs = 0.10, CI95% [0.06–0.15], p = 0.000013) and also between the robustness index (HRCD) and SR (rs = 0.06, CI95% [0.02–0.11], p = 0.007).

Analysis of the classification and regression models using Random Forests resulted in mean significant, unbiased conditional R2c values of RCD, H and HRCD, with RCD3, H3 and SR between 0.10–0.18 (Table 2) and 0.14–0.16 (Table 3), 0.10–0.15 (Table 4), respectively. Furthermore, the predictor variable “seed provenance” (23 provenances) also significantly explained the variation in SR, RCD3 and H3 (unbiased conditional R2c = 0.06–0.17). By contrast, the variables “tree species and hybrids” (10 classes) and “field trial” (two classes) were the predictor variables that explained the least amount of variation in these three response variables. Besides, the predictor variable “seedling of pure or hybrid parents” (two classes) was unable to explain the variation of SR, RCD3 and H3. The biased marginal values R2m often differed greatly from R2c (Tables 2–4).

Table 2 Unbiased conditional R2c values and their 95% confidence interval (CI95%) obtained by bootstrapping (10,000 iterations) of each fixed predictor.

Unbiased conditional R2c values and their 95% confidence interval (CI95%) obtained by bootstrapping (10,000 iterations) of each fixed predictor (root collar diameter of seedling (RCD), seedling height (H) and seedling robustness index (HRCD) before planting in 2018), analyzing “tree species and hybrids” (10 classes) (called species), “seed provenance” (23 classes) (called provenance) and “field trial” (two classes) with respect to root collar diameter of seedling after 44 months of planting in both field trials (RCD3) when all other predictor variables in the Random Forest regression models were considered together; SD = Standard Deviation of R2c; R2m = Biased marginal R2 of RCD3 with respect to each fixed predictor; bold = mean R2c values of RCD, H and HRCD.

Fixed predictor	Mean unbiased R2c
(predictor vs. RCD3)	SD	CI 95%	Biased R2m
(predictor vs. RCD3)	
H	0.18	0.00	0.18	0.19	0.0	
RCD	0.11	0.00	0.11	0.12	0.0	
Provenance	0.10	0.01	0.09	0.11	0.20	
Species	0.03	0.01	0.01	0.04	0.21	
Field trial	0.0				0.02	
Fixed predictor						
RCD	0.16	0.00	0.16	0.17	0.0	
Provenance	0.09	0.01	0.08	0.10	0.20	
Species	0.03	0.01	0.02	0.04	0.21	
Field trial	0.0				0.02	
Fixed predictor					
H	0.10	0.00	0.09	0.10	0	
Provenance	0.08	0.00	0.08	0.09	0.20	
Species	0.04	0.00	0.03	0.05	0.21	
Field trial	0.0				0.02	
Fixed predictor						
HRCD	0.16	0.01	0.16	0.17	0	
Provenance	0.08	0.01	0.08	0.09	0.20	
Species	0.02	0.01	0.01	0.03	0.21	
Field trial	0.0				0.02	
Note:

Since RCD and H were not strongly correlated with each other (rs < 0.7), but HRCD was often strongly associated with RCD or H, the RF models were constructed with both RCD and H and the other three variables together (“tree species and their hybrids”, “seed provenance” and “field trial”) but without HRCD and then, also with either RCD, H or HRCD alone and the other three variables.

Table 3 Unbiased conditional R2c values and their 95% confidence interval (CI95%) by bootstrap (10,000 iterations) for predictor variables.

Unbiased conditional R2c values and their 95% confidence interval (CI95%) by bootstrap (10,000 iterations) for the predictor variables root collar diameter of seedling (RCD), seedling height (H) and seedling robustness index (HRCD) before planting in 2018, analyzing “tree species and hybrids” (10 classes) (called species), “seed provenance” (23 classes) (called provenance) and “field trial” (two classes) with respect to seedling height 44 months after planting in both field trials (H3) when all other predictor variables in the Random Forest regression models were considered together. SD = Standard Deviation of R2c; R2m = Biased marginal R2 of H3 with respect to each fixed predictor; bold = mean R2c values of RCD, H and HD.

Fixed predictor	Mean unbiased R2c
(predictor vs. H3)	SD	CI 95%	Biased R2m
(predictor vs. H3)	
Provenance	0.17	0.00	0.16	0.17	0.41	
RCD	0.15	0.01	0.14	0.16	0.0	
H	0.14	0.01	0.13	0.15	0.13	
Species	0.06	0.01	0.04	0.08	0.41	
Field trial	0.0				0.0	
Fixed predictor					
Provenance	0.16	0.01	0.14	0.17	0.41	
RCD	0.14	0.00	0.13	0.14	0.0	
Species	0.07	0.01	0.06	0.09	0.41	
Field trial	0.0				0.0	
Fixed predictor					
H	0.16	0.01	0.15	0.16	0.13	
Provenance	0.15	0.01	0.14	0.16	0.41	
Species	0.07	0.01	0.05	0.09	0.41	
Field trial	0.0				0.0	
Fixed predictor					
Provenance	0.15	0.01	0.14	0.15	0.41	
HRCD	0.14	0.01	0.13	0.15	0.0	
Species	0.06	0.01	0.04	0.08	0.41	
Field trial	0.0				0.0	
Note:

Since RCD and H were not strongly correlated with each other (rs < 0.7), but HRCD was often strongly associated with RCD or H, the RF models were constructed with both RCD and H and the other three variables together (“tree species and their hybrids”, “seed provenance” and “field trial”) but without HRCD and then, also with either RCD, H or HRCD alone and the other three variables.

Table 4 Unbiased conditional R2c values and their 95% confidence interval (CI95%) obtained by bootstrapping (10,000 iterations) of each fixed predictor.

Unbiased conditional R2c values and their 95% confidence interval (CI95%) obtained by bootstrapping (10,000 iterations) of each fixed predictor (root collar diameter of seedling (RCD), seedling height (H) and seedling robustness index (HRCD) before planting in 2018), analyzing “tree species and hybrids” (10 classes) (called species), “seed provenance” (23 classes) (called provenance) and “field trial” (two classes) with respect to survival rate (SR) of the seedlings after 44 months of planting in the two field trials, when all other predictor variables in the Random Forest classification models were considered together; SD = Standard Deviation of R2c; R2m = Biased marginal R2 of SR with respect to each fixed predictor; bold = mean R2c values of RCD, H and HRCD.

Fixed predictor	Mean unbiased R2c
(predictor vs. SR)	SD	CI 95%	Biased R2m
(predictor vs. SR)	
RCD	0.15	0.00	0.15	0.16	0.59	
H	0.14	0.00	0.13	0.15	0.58	
Provenance	0.06	0.00	0.06	0.07	0.55	
Species	0.0				0.54	
Field trial	0.0				0.65	
Fixed predictor					
Provenance	0.12	0.01	0.11	0.13	0.55	
RCD	0.12	0.01	0.11	0.13	0.59	
Species	0.11	0.01	0.09	0.12	0.54	
Field trial	0.09	0.01	0.07	0.11	0.65	
Fixed predictor					
Provenance	0.10	0.00	0.10	0.11	0.55	
H	0.10	0.00	0.10	0.11	0.58	
Field trial	0.10	0.01	0.10	0.11	0.65	
Species	0.09	0.00	0.09	0.10	0.54	
Fixed predictor					
Provenance	0.13	0.01	0.12	0.14	0.55	
HRCD	0.13	0.01	0.12	0.14	0.73	
Species	0.12	0.01	0.11	0.13	0.54	
Field trial	0.11	0.01	0.10	0.13	0.65	
Note:

Since RCD and H were not strongly correlated with each other (rs < 0.7), but HRCD was often strongly associated with RCD or H, the RF models were constructed with both RCD and H and the other three variables together (“tree species and their hybrids”, “seed provenance” and “field trial”) but without HRCD and then, also with either RCD, H or HRCD alone and the other three variables.

The GLM resulted in mean significant, although very low unbiased conditional R2c values of RCD, H and HRCD, with RCD3 and H3 between 0–0.03, respectively. Moreover, the predictor “seed provenance” significantly contributed to explain the variation in RCD3 and H3 (R2c = 0.03 and 0.05; (CI95% [0.01–0.05] and CI95% [0.03–0.07])). Finally, the GLM predictor variable “field trial” significantly contributed to explain the variation in SR with R2c equal 0.09 (CI95% [0.06–0.12]).

When grouped in classes of 2 mm, 2 cm and 0.3 cm/mm2 width, the values of RCD, H and HRCD of the 15-month seedlings (before planting) affected RCD3, H3 and SR 44 months after planting in both field trials. On average, we observed positive relationships between RCD before planting and 44 months later, between H before planting and 44 months after planting, and between both H and HRCD before planting and SR 44 months after planting, but not between RCD and SR (Figs. 2–3).

Figure 2 Relationships between (A) seedling root collar diameter classes (2 mm class width) and (B) seedling height classes (2 cm class width) before planting and the same variables 44 months after planting.

Relationships between (A) seedling root collar diameter (RCD) classes (2 mm class width) and (B) seedling height (H) classes (2 cm class width) before planting (at 15 months old) and the same variables 44 months (3 years) after planting in two field trials, across all pure species and their hybrids, showing Mean H (hybrids), Mean P (pure species) and groups of pine species and their hybrids; PA‐P = Pinus arizonica pure species, PD‐P = P. durangensis pure species, PE‐P = P. engelmannii pure species, PL‐P = P. leiophylla pure species, and PT‐P = P. teocote pure species; PA‐H = hybrids of Pinus arizonica × P. durangensis genetically more similar to P. arizonica; PD‐H = hybrids of P. durangensis × P. arizonica and P. durangensis × P. engelmannii both genetically more similar to P. durangensis; PE‐H = hybrids of P. engelmannii × P. arizonica genetically more similar to P. engelmannii; PL‐H = hybrids of P. leiophylla × P. teocote genetically more similar to P. leiophylla; PT‐H = P. leiophylla × P. teocote genetically more similar to P. teocote (for details, see Hernández-Velasco et al., 2021; Sánchez-Hernández et al., 2022).

Figure 3 Relationships between (A) seedling root collar diameter classes, (B) seedling height classes, (C) seedling robustness index classes before planting and survival rate 44 months after planting.

Relationships between (A) seedling root collar diameter (RCD) classes (2 mm class width), (B) seedling height (H) classes (2 cm class width), (C) seedling robustness index (HRCD) classes (0.3 cm/mm2 class width) before planting (at 15 months old) and survival rate 44 months after planting in two field trials, averaged and grouped by pine species and their hybrids (abbreviations are shown in Fig. 1).

The low width of the CI95% of the study results, especially in correlations, shows that the trial fields had environmental conditions quite homogeneous (Table S3; Tables 2–4).

Discussion

In accordance with our hypothesis, the results indicate that the seedling root collar diameter (RCD) and height (H) of 15-month-old seedlings of the pure species Pinus arizonica, P. durangensis, P. engelmannii, P. leiophylla and P. teocote, as well as their hybrids, in several cases had a significant and positive (Fig. 1, Table S3), although weak effect on the same respective dimensions and on the survival rate (SR) 44 months after planting (explained between 8% and 18% of the variance in the plants’ corresponding dimensions and explained between 12% and 15% of the survival rate based on Random Forests models; Tables 2–4, Figs. 2–3). In both field trials, consequently, the advantages of large over small seedlings were observed, similar to Garau et al. (2009), Landis & Dumroese (2010) and Grossnickle & MacDonald (2018).

Two important factors to explain this low association between the pre-planting pine seedling dimensions and the corresponding dimensions and survival rates after 44 months in the trial fields could be the strong damage observed caused by pocket gopher, Thomomys spp. and the typically low winter-spring precipitation in both field trials, which was particularly low in 2019 and 2021 (<100 mm in each year) (CONAGUA, 2024). In practice, larger pine seedlings may be more susceptible to damage from the burrowing activities of Thomomys spp., as their root systems could be disrupted by tunnels (Knight, 2000), and they could also be more attractive to gophers due to their size and higher nutrient content (Hunt, 1992). Besides, larger seedlings have a larger transpiration demand, which may have a negative impact on their growth and survival during periods of drought or in particularly dry locations. Therefore, larger seedlings may suffer more from water stress compared to smaller seedlings (Oliet et al., 2019).

As in our case, Li et al. (2011) in a 3-year study and Tsakaldimi, Ganatsas & Jacobs (2013) in a 2-year study also found positive, but stronger correlations of the same variables, probably caused by their reported active intervention in the trail fields (Li et al., 2011) and by the shorter study time (Tsakaldimi, Ganatsas & Jacobs, 2013). Dey & Parker (1997) described similar results in red oak (Quercus rubra L.) seedlings underplanted in a central Ontario shelterwood. Li et al. (2011) reported similar findings for Larix olgensis seedlings. Alike to the observations of Li et al. (2011) and consistent with our assumption, we hardly find correlation between robustness index (HRCD) before planting and the plants dimensions 44 months after planting, with the only exception of a weak negative, although significant correlation with the seedlings diameter (Fig. 1, Table S3). Prieto-Ruíz et al. (2018) studied the seedlings behavior of Mexican Pinus cooperi 13 months after planting, comparing seedlings with two initial morphological conditions (RCD 3.5 mm and H 10–14 cm vs. RCD 4 mm and H 15–25 cm), and found that the seedling size did not influence survival, but higher values of RCD and H favored growth.

In reforestation practices, many traits and drivers have been identified and proposed as important factors for the near-optimal state of forest plantations (Prieto-Ruíz et al., 2016; Grossnickle & MacDonald, 2018). Nevertheless, the results regarding the assessment of the predictors evaluated in this study suggest that other variables, such as seed provenance (Tables S1–S2 and Tables 2–4), were more important than the initial seedling dimensions for the survival and plants’ size after 44 months in the field. According to our results regarding the width of CI95%, the variation in microhabitat conditions slightly influenced the magnitude of variable importance, but not the ranking of importance (Table S3, Tables 2–4).

Given that the average seedlings per each seed provenance were grown under similar conditions of temperature and humidity, both in the greenhouse and after planting, any initial differences in seedlings size and also 44 months later in the field trials, as well as differences in survival rates can be partly attributed to genetics and maternal effects. The influence of two genetic variables (“seed provenance” and “tree species and hybrids”) was taken into account in this study, Tables S1–S2 and 2–4).

Thus, the weak correlation between nursery seedlings size before and after 44 months of planting could also be due to adaptation factors. Since, considering the climatic conditions of each of the 23 seed provenances, some seedling genotypes may have grown better under the tropical humid, nutrient-rich greenhouse conditions of the nursery (Hernández-Velasco et al., 2021) than if they had grown in the natural colder and drier environment where the respective mother tree lives. However, after the nursery stage, when the same seedlings were re-transplanted to the colder, drier and nutrient-poorer environment of the trial fields (Sánchez-Hernández et al., 2022), the mortality rate of large seedlings may have been greater than that in shorter seedlings. Large seedlings could be less genetically adapted to the planting site by epigenetic memory effects (Gömöry et al., 2017). The highly artificial nursery conditions could also have altered genotype selection (e.g., homozygote excess) in the seedlings (Gillet, Ziehe & Gregorius, 2016) which may have selectively influenced the survival of individual genotypes after planting (Campbell & Sorensen, 1984).

The comparison of the initial development of the RCD and H of our test seedlings with the current Mexican standard NMX-AA-170-SCFI-2016 (Secretaría de Economía, 2016) shows that the RCD almost always exceeded the standard minimum sizes after a period indicated for each species in nursery conditions. However, the majority of the seedlings in our study did not reach the minimum standard’s H in such period. This also means that most of these seedlings initially had a better (higher) robustness index (HRCD) than the aforementioned mentioned standard suggests.

Our results also show that the Random Forests models (RF) detected a higher unbiased conditional R2c than the generalized linear models (GLM) used. RF overfitting was unlikely, due to the very small confidence intervals found with such models (Tables 2–4). These differences may be attributed to a larger robustness, flexibility and to the ability of RF to capture complex patterns in the data (Breiman, 2001). On the other hand, the flexibility of GLM is limited because these models are usually linear with respect to the predictors, even if they can model nonlinear relationships through appropriate link functions and polynomial or interactive terms (Hardin & Hilbe, 2007).

The large difference found between the squared Spearman correlation coefficient (rs2) of individual variables, biased marginal R2m and the unbiased conditional R2c of classification and regression models (Table S3 vs. Tables 2–4), which no longer included the influence of the other specific predictors variables (also referred to here as “confounding variables”), highlights the importance of calculating R2c in order to mitigate the effects of the studied confounding variables. However, R2c may change if more confounding variables are added to the study.

Conclusion

In this study, the effects of initial morphological attributes such as seedling diameter, height, and robustness index were significant, although in general not good predictors of growth and survival 44 months after planting those seedlings in field trials. Moreover, the genetic variables “seed provenance” and “species affiliation” were just as important as the seedling dimension for growth and survival. Consequently, the adequate selection of seed origins and tree species and their hybrids might be an important factor for the success of reforestation in the Sierra Madre Occidental.

Since the experiment was conducted using only regional provenances in just two trial fields, the representativeness of the results is limited. However, our findings offer preliminary insights that could help refine standards for plant nurseries seeking to enhance survival and growth rates in reforestations. Specifically, our results suggest that the Mexican standard NMX-AA-170-SCFI-2016 should be revised to incorporate genetic factors and consider not only seedlings dimensions but also the climatic conditions of the seed provenance and the site of final plantation. Further research is needed to address these gaps in knowledge.

Supplemental Information

Supplemental Information 1 Number of seedlings (N) of the five pure (-P) Pinus species and their seed provenances in the study, in each separate field trial and in both trials together, 15 months after sowing in the nursery.

Supplemental Information 2 Number (N) of hybrid seedlings (-H) of the five Pinus species and their seed provenances in the study, in each separate field trial and in both trials together, 15 months after sowing in the nursery.

Supplemental Information 3 Significant correlation between the root collar diameter, height, robustness index of seedlings before planting in 2018 and RCD, H and the survival rate of the same seedlings after planting.

Significant correlation (rs) between the root collar diameter (RCD, mm), height (H, cm), robustness index (HRCD, cm/mm2) of seedlings before planting in 2018 and RCD3, H3 and the survival rate (SR) of the same seedlings 44 months after planting in the field trials, for each pine pure species (−P) and their hybrids (−H), and their 95% confidence interval (CI95%) by bootstrap (1,000,000 iterations).

Supplemental Information 4 Raw data.

We are thankful to Ricardo Silas Sánchez Hernández† and Carlos Adrián Vásquez Berumen for their assistance in data collection.

Additional Information and Declarations

Competing Interests

Author Contributions

Data Availability

Christian Wehenkel is an Academic Editor for PeerJ.

José Alberto Ponce-Figueroa performed the experiments, analyzed the data, authored or reviewed drafts of the article, and approved the final draft.

Pablo Antúnez analyzed the data, prepared figures and/or tables, authored or reviewed drafts of the article, and approved the final draft.

José Ciro Hernández-Díaz analyzed the data, authored or reviewed drafts of the article, and approved the final draft.

José Ángel Prieto-Ruíz analyzed the data, authored or reviewed drafts of the article, and approved the final draft.

Artemio Carrillo-Parra analyzed the data, authored or reviewed drafts of the article, and approved the final draft.

Pablito Marcelo López-Serrano analyzed the data, authored or reviewed drafts of the article, and approved the final draft.

Christian Wehenkel conceived and designed the experiments, performed the experiments, analyzed the data, prepared figures and/or tables, authored or reviewed drafts of the article, and approved the final draft.

The following information was supplied regarding data availability:

The raw data is available in the Supplemental File.

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
