# Peer review of "Effect of seedling size on post-planting growth and survival of five Mexican Pinus species and their hybrids"

_PeerJ, doi:10.7717/peerj.18725_

## Round 0.1 · original submission · Major Revisions

Dear Authors,

Two independent reviewers have concluded their revision of the manuscript. Based on their opinion, many flaws have been evidenced that preclude the acceptance of the manuscript in its present form, and I agree with their statement.

I suggest the authors to respond the criticisms replaying point-to point to reviewer' comments after carried out the revision of the manuscript.

·

Basic reporting

I have received for review a manuscript titled "Effect of seedling size on post-planting growth and survival of five Mexican Pinus species and their hybrids." The main aim of the study was to evaluate the impact of morphological traits of seedlings of five pine species and their hybrids on survival after 44 months of growth. The subject matter is important for the area discussed in the manuscript, which faces significant deforestation challenges. Additionally, the newly established forests in the selected research area largely fail to thrive, making the research question posed by the authors crucial for both local science and forestry management.

The publication is written in clear and coherent English, although I must highlight that I am not a native speaker, so my assessment in this regard may be limited. The manuscript contains appropriate figures and tables, which are correctly described.

However, I would like to point out that the scientific and research problem discussed in the manuscript is neither new nor innovative, as similar topics have been addressed in works such as Prieto-Ruíz et al. (2016) and MacDonald (2018). Therefore, my initial conclusion is that the reviewed manuscript does not introduce novel topics or research to the field but rather continues a research pattern that has been in use for some time.

Experimental design

The research question that the authors decided to test in the manuscript is presented at the end of the Introduction section, which is typical for this type of text. Unfortunately, I must point out that it is formulated without a thorough logical analysis of the problem. As a result, a reader familiar with the research topic is left with a sense of dissatisfaction, as the authors do not even attempt to explore beyond the presentation of an obvious research need. In my view, manuscripts submitted to journals of the proposed calibre should describe the scientific problem in more detail, formulating hypotheses. I therefore suggest revising the main objective and attempting to formulate specific hypotheses, which would allow readers to understand the broader context of the research.

The research was conducted on plants, which means that ethical considerations can be omitted from this review. However, I would like to focus on the experimental conditions under which the study was conducted. The presented study involved two field trials, Mesa Alta and Mesa Seca, and a significant number of seedlings. The sample size was sufficient, and the experimental design was appropriate. My concerns focus on a few elements: while two field trials are adequate for pilot studies, they may not be sufficient to provide a broader perspective. Additionally, no microhabitat analyses were conducted, despite the fact that local microhabitat variables can have a dominant influence on the obtained results. I therefore recommend incorporating into the presented statistical analyses models, such as those proposed by Munoz and Sanchez (2019), which account for microhabitat diversity, and I suggest that the manuscript be supplemented with the results of these analyses.

Validity of the findings

The statistical analysis presented utilises appropriate analytical tools, and the results are displayed with due care. The tables and figures are correctly labelled. However, several gaps must be addressed: the authors used Spearman’s rank correlation coefficient (rs) to examine correlations between variables such as root collar diameter (RCD), height (H), and robustness index (HRCD). While Spearman’s correlation is a valid tool for investigating non-linear relationships between variable ranks, its application in this case may be limited. Spearman’s coefficient measures monotonicity rather than linearity. The authors did not consider other, more advanced modelling techniques (e.g. non-linear models or multivariate analysis methods), which could potentially capture more complex relationships between variables.
The Generalised Linear Model (GLM) employed is a classical modelling tool, but its restriction to modelling primarily linear relationships can be problematic when studying complex ecological phenomena. The low R²c values obtained for GLM (ranging from 0 to 0.03) confirm that this model is insufficient to adequately describe the dynamics of seedling growth. It is therefore recommended that more sophisticated models, such as Generalised Additive Mixed Models (GAMMs) or Bayesian models, be used in the presented statistical analysis. These models are more effective at capturing the complexity of ecological research, including studies like the one presented in this manuscript.

The results obtained by the authors demonstrated the significance of “seed provenance” as a predictor, while variables such as “species,” “hybridisation,” or “field trial” had less influence on the response variables. Unfortunately, I found no thorough assessment in the text of the reasons behind these findings. In my view, the impact of microhabitat conditions was likely the most crucial factor. Therefore, a need to supplement the manuscript with additional calculations.

Additional comments

The review presented above aims to highlight the subjective weaknesses of the manuscript. However, I believe that the material presented is a solid piece of scientific work and, despite its shortcomings – particularly in the aspect of statistical analysis – it should be considered for publication after a major revision and a subsequent re-review of the manuscript.

Reviewer 2 ·

Basic reporting

This article focuses on the effect of seedling vigor and size on seedling adaptation. It further explores the significance of various local Mexican trees and their hybrid seedlings in restoring deforestation across Mexico in the future. The article is well-written, but I suggest some minor revisions before acceptance.
The abstract should be revised by reducing the background and discussion to 2-4 lines and expanding the methods and results sections.
The introduction is too lengthy and should be condensed to 1.5 pages by removing unnecessary information.
In line 74, please provide additional explanation for 'speed of root-soil contact.'
In line 133, consider including the seedling soil nutrient composition, such as N%, P%, etc.
Table 1 and Table 2 should be merged into a single table.
The results of Table 3 should be presented as a figure rather than a table.

After addressing the revisions mentioned above, I endorse the publication of this article

Experimental design

The current article aligns well with the aims and scope of the journal. The methodology is clearly written and explained in sufficient detail

Validity of the findings

The authors have provided complete statistical data and conclusions are written well and supporting the results.

---

## Round 0.2 · accepted · Accept

The authors have carried out all requested changes to the manuscript and now it is significantly improved and worthy of publication.

·

Basic reporting

The reviewed manuscript titled "Effect of seedling size on post-planting growth and survival of five Mexican Pinus species and their hybrids" is a solid scientific work. Particularly after incorporating all the suggestions and corrections proposed in previous reviews, it meets the standards of the journal PeerJ. The language used is correct and clear, while the tables and figures are appropriately aligned with the context of the publication. The proposed literature review fulfills expectations and is well-suited to the topic of the article.

However, it should be noted that the research problem addressed in the manuscript is neither new nor particularly innovative. The text serves more as an extension of the current research agenda rather than a groundbreaking contribution to the field.

Experimental design

After incorporating the feedback from previous reviews and adding the necessary calculations along with the resulting information to the manuscript, no further corrections are required. As a reviewer, I have no additional comments regarding the manuscript.

Validity of the findings

The reviewed manuscript meets the expectations for this type of scientific work in terms of the experimental design. Although conducting the study on only two research sites may raise some concerns, the addition of an appropriate explanation in lines 359–360 sufficiently and transparently addresses this issue. Furthermore, in my opinion, the inclusion of additional statistical analyses has significantly enhanced the manuscript's scientific value.

Additional comments

The resubmitted manuscript incorporates most of the previously suggested corrections. Therefore, I believe that its current version is sufficient to recommend it for publication in the scientific journal selected by the authors.

Reviewer 2 ·

Basic reporting

I have reviewed the revised manuscript, and the authors have effectively addressed all my comments. I endorse the acceptance of this manuscript in its current form.

Experimental design

The manuscript has been revised in accordance with my comments.

Validity of the findings

N/A

Additional comments

N/A